# Intestinal Epithelial STAT6 Activation Rescues the Defective Anti-Helminth Responses Caused by *Ogt* Deletion

**DOI:** 10.3390/ijms231911137

**Published:** 2022-09-22

**Authors:** Xiwen Xiong, Rong Huang, Zun Li, Chenyan Yang, Qingzhi Wang, Hai-Bin Ruan, Lin Xu

**Affiliations:** 1School of Forensic Medicine, Xinxiang Medical University, Xinxiang 453000, China; 2School of Laboratory Medicine, Xinxiang Medical University, Xinxiang 453000, China; 3Department of Integrative Biology and Physiology, University of Minnesota Medical School, Minneapolis, MN 55455, USA

**Keywords:** OGT, tuft cell, goblet cell, helminth, STAT6, type 2 immunity

## Abstract

Dynamic regulation of intestinal epithelial cell (IEC) proliferation and differentiation is crucial for maintaining mucosa homeostasis and the response to helminth infection. O-GlcNAc transferase (OGT), an enzyme catalyzing the transfer of GlcNAc from the donor substrate UDP-GlcNAc onto acceptor proteins, has been proposed to promote intestinal epithelial remodeling for helminth expulsion by modifying and activating epithelial STAT6, but whether the IEC intrinsic OGT-STAT6 axis is involved in anti-helminth responses has not been tested in vivo. Here, we show that the inducible deletion of *Ogt* in IECs of adult mice leads to reduced tuft and goblet cell differentiation, increased crypt cell proliferation, and aberrant Paneth cell localization. By using a mouse model with concurrent *Ogt* deletion and STAT6 overexpression in IECs, we provide direct in vivo evidence that STAT6 acts downstream of OGT to control tuft and goblet cell differentiation in IECs. However, epithelial OGT regulates crypt cell proliferation and Paneth cell differentiation in a STAT6-independent pathway. Our results verify that protein O-GlcNAcylation in IECs is crucial for maintaining epithelial homeostasis and anti-helminthic type 2 immune responses.

## 1. Introduction

The intestinal epithelium consists of a single layer of columnar intestinal epithelial cells (IECs) lining the luminal surface, and acts as a physical and immunological barrier between the host and its luminal contents [1,2]. IECs are composed of several cell types including proliferative intestinal stem cells (ISCs) and progenitor cells (also named transit amplifying cells), as well as differentiated absorptive enterocytes and secretory Paneth, goblet, tuft, and enteroendocrine cells (EECs) [3]. ISCs divide to self-renew and generate more rapidly proliferating progenitors that give rise to all the other IEC types and facilitate the continuous homeostatic renewal of the intestinal mucosal epithelium [4].

Tuft and goblet cells constitute only a minor fraction of the mouse intestinal epithelium, but strict control of their differentiation is critical for driving type 2 mucosal immune responses to parasitic helminth infection [2,5]. After helminth infection, IL-25 is produced by tuft cells, which further activates group 2 innate lymphoid cells (ILC2s) to secrete IL-13, which acts on crypt progenitor cells to promote the differentiation of tuft and goblet cells. The tuft cell-ILC2 feedforward circuit potentiates epithelial remodeling and mobilizes a broad range of type 2 immunity-induced events such as smooth muscle hypercontractility, mucus production, and luminal fluidity, to expel parasites [6,7,8]. STAT6 functions as a transcription factor to mediate type 2 immune responses following IL-4/IL-13 activation, playing critical roles in intestinal helminth expulsion and allergic disorders [9,10]. Whole-body *Stat6*^*−/−*^ mice show an abrogated expansion of tuft and goblet cells and compromised worm clearance after helminth infection, while IEC-specific STAT6 activation promotes the differentiation of tuft and goblet cells, and protects against helminths, indicating that epithelial STAT6 is necessary and sufficient to regulate tuft and goblet cell differentiation in IECs [6,11,12,13].

O-GlcNAcylation is a type of post-translational modification consisting of the addition of a single N-acetyl-glucosamine molecule (GlcNAc) to the specific serine/threonine residues of proteins [14]. Intriguingly, O-GlcNAc cycling is catalyzed and removed by a single pair of enzymes, O-GlcNAc transferase (OGT) and O-GlcNAcase (OGA), respectively [15]. Mounting evidence has shown that O-GlcNAcylation serves as a nutrient and stress sensor regulating biological processes, with its effects ranging from transcription and translation to signal transduction and metabolism [16]. We recently revealed that Vil-Cre-mediated *Ogt* deletion led to blunted tuft and goblet cell hyperplasia after helminth infection. Defective STAT6 O-GlcNAcylation and activation in *Ogt* knockout IECs is postulated as a key mechanism responsible for impaired epithelial remodeling and anti-helminth immunity [12]. However, we still lack direct animal study evidence to verify the role of the OGT-STAT6 axis in regulating tuft and goblet cell differentiation. In addition, since the *Vil1* gene is expressed during embryogenesis, Vil-Cre-mediated *Ogt* deletion in mice resulted in severe intestinal damage and premature death at approximately 6–8 weeks of age, which may confound the findings regarding IEC differentiation [17,18]. In the present study, by using inducible *Ogt* IEC-specific knockout and STAT6 IEC-specific overexpression transgenic mouse models, we provide direct in vivo evidence that epithelial STAT6 is a downstream mediator of OGT to regulate tuft and goblet cell differentiation, and anti-helminth responses in adult mice.

## 2. Results

### 2.1. Generation of Inducible IEC-Specific Ogt Knockout Mice

To circumvent the severe intestinal damage and premature lethality in Vil-Cre-mediated *Ogt* knockout mice, we established a mouse model with epithelial *Ogt* ablation in adult mice by crossing *Ogt* floxed mice with Vil-CreERT2 mice. We treated *Ogt* floxed; Vil-CreERT2 mice with tamoxifen (TAM) for 3 consecutive days to activate Cre recombinase. Over a 10-day period following the end of TAM treatment, *Ogt* and global O-GlcNAcylation were efficiently deleted in IECs (Figure 1A–D; hereafter, TAM treated *Ogt* floxed; Vil-CreERT2 mice are referred as *Ogt^i^**^∆IEC^*, whereas TAM treated *Ogt* floxed mice are referred to as *Ogt^Ctrl^* mice). Intriguingly, both male and female *Ogt^i^**^∆IEC^* mice appeared normal; however, they became substantially lighter in body weight on day 10 after *Ogt* deletion (Figure 1E).

### 2.2. Deletion of Ogt in IECs of Adult Mice Leads to Abnormal Epithelial Architecture

Notably, the small intestine of *Ogt^i^**^∆IEC^* mice exhibited structural changes during histological analysis (H&E) on day 10 post TAM treatment (Figure 2A). The crypt depth significantly increased by ~30% and the villus also became slightly elongated by ~10% (Figure 2B). Consistent with the enlarged intestinal crypt phenotype, Ki67 immunostaining revealed that *Ogt^i^**^∆IEC^* mice had more proliferative cells in the crypt than *Ogt^Ctrl^* mice (Figure 2C,D). We previously revealed that Vil-Cre-mediated *Ogt* knockout resulted in the reduced abundance of Paneth cells in the crypt [18]. Intriguingly, *Ogt^Ctrl^* and *Ogt^i^**^∆IEC^* mice had comparable numbers of intestinal Paneth cells, as illustrated by lysozyme immunostaining (Figure 2E,F). Lysozyme-positive Paneth cells in *Ogt^Ctrl^* mice were located at the bottom of the crypt. Surprisingly, Paneth cells in *Ogt^i^**^∆IEC^* mice were less uniform and not restricted to the bottom of the crypt (Figure 2E). Despite the aberrant location of Paneth cells, the intestinal mRNA levels of Paneth cell marker genes, i.e., *Lyz1* and *Defa6*, were not changed in *Ogt^i^**^∆IEC^* mice (Figure 2H). Importantly, we found no difference between the *Ogt^Ctrl^* and *Ogt^i^**^∆IEC^* groups in terms of the intestinal mRNA expression of enterocyte cell markers (*Fabp1*, *Slc5a1*) and ALP staining, indicating normal absorptive cell lineage differentiation in the absence of OGT (Figure 2G,H).

### 2.3. Epithelial OGT Is Required for Tuft and Goblet Cell Differentiation in Adult Mice

To investigate the functional outcome of *Ogt* ablation in tuft and goblet cell differentiation in adult mice, we performed qPCR to analyze the mRNA expression of tuft and goblet cell linage markers. Consistent with our previous findings in Vil-Cre-mediated *Ogt* knockout mice, we found that the expression of tuft (*Pou2f3*, *Trmp5*) and goblet (*Retnlb*, *Muc3*) cell marker genes was significantly reduced in IECs of *Ogt^i^**^∆IEC^* mice compared with *Ogt^Ctrl^* mice (Figure 3A). Furthermore, we assessed intestinal tuft and goblet cell abundance by performing DCLK1 immunostaining and Alcian blue staining, respectively. As expected, *Ogt^i^**^∆IEC^* mice exhibited a significant reduction in the number of tuft and goblet cells, suggesting OGT is required for intestinal tuft and goblet cell differentiation in adult mice (Figure 3B–E). To confirm whether STAT6 activity is impaired in IECs of *Ogt^i^**^∆IEC^* mice, we performed immunofluorescence and Western blot analyses to detect the STAT6 phosphorylation of tyrosine 641, which is activated by IL-4/IL-13 and is required for its nuclear translocation. Indeed, *Ogt^i^**^∆IEC^* mice exhibited significantly reduced epithelial STAT6 (Y641) phosphorylation, verifying that STAT6 activity in IECs is compromised in the absence of OGT (Figure 3F,G).

### 2.4. OGT Regulates Intestinal Crypt Cell Proliferation and Paneth Cell Differentiation in a STAT6-Independent Manner

Our previous and current data showed that *Ogt* deletion in IECs resulted in noticeably reduced STAT6 phosphorylation and activity [12]. To further explore whether enhanced STAT6 activity rescues the epithelial abnormalities caused by OGT deficiency, we generated a new mouse model with concurrent STAT6 overexpression and *Ogt* deletion in IECs. We first generated a transgenic mouse model in which a constitutively active form of STAT6 (V547A/T548A, STAT6vt) was expressed under the control of the *Vil1* gene promoter, named TgS6vt [19,20]. As shown in Figure 4A–D, TgS6vt mice exhibited a moderate epithelial expression of transgene STAT6vt (~7-fold increase) and reduced intestinal villus height and crypt depth compared with WT controls. We then crossed TgS6vt mice with *Ogt^i^**^∆IEC^* mice to produce *Ogt^i^**^∆IEC^*-TgS6vt mice, which specifically express STAT6vt in IECs with TAM-induced *Ogt* ablation. Importantly, when the intestinal villus height and crypt depth were measured, we observed a small but not significant decrease in villus height in *Ogt^i^**^∆IEC^*-TgS6vt mice. However, the crypt depth was comparable between *Ogt^i^**^∆IEC^* and *Ogt^i^**^∆IEC^*-TgS6vt groups (Figure 4E,F). Expectedly, Ki67 immunostaining revealed that the number of Ki67-positive proliferative cells in the crypt did not differ between the two genotypes, indicating that the activation of STAT6 in IECs has no effect on crypt cell proliferation in the absence of OGT (Figure 4G,H). In addition, STAT6vt overexpression failed to rescue the aberrant localization of Paneth cells caused by epithelial *Ogt* deletion, as illustrated by lysozyme immunostaining (Figure 4I,J).

### 2.5. STAT6 Activation Promotes Intestinal Tuft and Goblet Cell Differentiation in Ogt^i^^∆^^IEC^ Mice

Given that IEC intrinsic STAT6 is sufficient to drive intestinal secretory cell differentiation, we went on to examine whether epithelial STAT6 overexpression rescues the impaired tuft and goblet cell differentiation observed in *Ogt^i^**^∆IEC^* mice [13]. Indeed, *Ogt^i^**^∆IEC^*-TgS6vt mice had more tuft and goblet cells in the small intestine than *Ogt^i^**^∆IEC^* mice, as illustrated by DCLK1 and Alcian blue staining (Figure 5A,B,D,E). Accordingly, the intestinal expression of marker genes for tuft (*Pou2f3*, *Trpm5*) and goblet (*Retnlb*, *Muc3*) cells increased substantially in *Ogt^i^**^∆IEC^*-TgS6vt mice (Figure 5C,F).

### 2.6. STAT6 Overexpression Rescues the Defective Anti-Helminth Responses Caused by Epithelial Ogt Ablation

Since tuft and goblet cells are key components of type 2 immune responses to helminth infection, we infected *Ogt^i^**^∆IEC^* and *Ogt^i^**^∆IEC^*-TgS6vt mice with *H. poly*, a natural intestinal-dwelling parasitic helminth of mice [2,21]. Compared with *Ogt^i^**^∆IEC^* mice, *Ogt^i^**^∆IEC^*-TgS6vt mice displayed markedly amplified intestinal tuft and goblet cell hyperplasia after *H. poly* infection, indicating that the activation of IEC STAT6 is sufficient to promote anti-helminth epithelial remodeling in the absence of OGT (Figure 6A–D). These findings were verified by performing qPCR analysis of tuft and goblet cell markers (Figure 6E,F). As a result, although the number of adult *H. poly* worms colonized in the small intestine was comparable between the two genotypes, *Ogt^i^**^∆IEC^*-TgS6vt mice passed profoundly less worm eggs into feces than *Ogt^i^**^∆IEC^* mice, indicating that the overexpression of STAT6 in IECs rescues the impaired helminth expulsion caused by *Ogt* ablation (Figure 6G,H).

## 3. Discussion

By using Vil-Cre-mediated *Ogt* knockout (Vil-Ogt KO) mice, we previously demonstrated that O-GlcNAcylation is crucial in maintaining intestinal homeostasis [12,18,22]. Vil-Ogt KO mice had severe intestinal damages, including a disrupted epithelial barrier, microbial dysbiosis, Paneth cell dysfunction, and intestinal inflammation [18]. However, Vil-Cre-mediated recombination occurs at embryonic day 9 in the visceral endoderm [17]. Notably, OGT is essential for embryonic stem cell viability and mouse ontogeny [23]. Therefore, we speculate that the high premature mortality in Vil-Ogt KO mice is probably due to some unknown developmental defects caused by *Ogt* ablation at the embryonic stage. The severe intestinal damages and high premature mortality in Vil-Ogt KO mice may confound our findings regarding IEC differentiation and obstruct us to further study the role of OGT in regulating epithelial homeostasis in the adult stage. In the current study, we trigged epithelial *Ogt* knockout (*Ogt^i^**^∆IEC^*) via the use of TAM-inducible Cre (Vil-CreER) to circumvent these abnormalities. *Ogt^i^**^∆IEC^* mice were healthy after TAM administration; however, they had an approximately 5% lower average body weight than *Ogt^Ctrl^* mice. Importantly, despite an abnormal epithelium architecture (i.e., expanded crypts and slightly longer villi), *Ogt^i^**^∆IEC^* mice did not develop defective barrier functions and spontaneous inflammation in the intestine. Thus, these data suggest that intrinsic IEC *Ogt* deletion in adult mice is not sufficient to cause spontaneous intestinal inflammation and a high risk of mortality.

Tuft cell-derived IL-25 triggers an intestinal ILC2-epithelial response circuit for anti-helminth activity. We previously found that although intestinal tuft abundance was reduced, the Vil-Ogt KO mice had a similar frequency of ILC2s in the lamina propria (LP) at the naïve state. Moreover, it must be pointed out, as reported by Schubart et al., that LP ILC2s were not increased in the naïve mice with IEC-specific STAT6vt overexpression although these mice had more intestinal tuft cells [13]. These results indicate that the alteration of intestinal tuft cell abundance alone is not sufficient to influence the accumulation of ILC2s in the LP. However, we speculate, under the circumstances of *H. poly* infection, that increased tuft cell abundance may lead to the activation of ILC2s and other types of immune cells to promote type 2 immune responses for helminth expulsion. Indeed, we observed that Vil-Ogt KO mice were unable to generate a functional type 2 immune response for helminth expulsion due to impaired tuft and goblet cell differentiation [12].

We found that STAT6 itself could be O-GlcNAc modified, and this O-GlcNAcylation was indispensable for the proper activation of STAT6 by IL-4/IL-13 [12]. It is known that IL-4/IL-13-mediated STAT6 signaling plays a central role in promoting tuft and goblet cell hyperplasia in the small intestine during helminth infection [24,25]. *Stat6^−/−^* mice exhibit blunted tuft and goblet cell hyperplasia and type 2 immune responses upon helminth infection [6,11,12]. In contrast, the specific expression of STAT6 in IECs promotes secretory cell differentiation (including tuft and goblet cells) and protects against helminth infection [13,20]. Thus, we hypothesized that *Ogt* knockout in IECs ablated STAT6 O-GlcNAcylation and reduced its transcriptional activity, thereby suppressing helminth-induced intestinal epithelial remodeling. To verify that STAT6 acts downstream of OGT to control helminth infection-induced epithelial hyperplasia, we generated *Ogt^i^**^∆IEC^*-TgS6vt mice with concurrent STAT6vt overexpression and *Ogt* deletion in IECs. STAT6vt is a constitutively activated form of STAT6 with two amino acid changes in the SH2 domain (V547A/T548A), resulting in JAK-independent Tyr641 phosphorylation and activation [19]. Our data clearly support our hypothesis that STAT6 mediates OGT’s function in regulating mucosal type 2 immune responses against helminth infection.

As we mentioned above, Schubart et al. showed that STAT6vt overexpression in IECs not only promoted tuft and goblet cell differentiation but also induced Paneth cell expansion [13]. Even though TgS6vt mice had more intestinal tuft and goblet cells, we did not observe noticeable differences in the number of Paneth cells between WT and TgS6vt groups. The reason for the discrepancy regarding Paneth cell differentiation between these two mouse models is still unclear and could be explained by the complex interplay of STAT6 with other unknown factors, i.e., microbiota-derived metabolites and signals, to regulate Paneth cell differentiation. Again, our data showed that IEC-specific STAT6vt overexpression did not alter the number nor localization of Paneth cells in the small intestine of *Ogt^i^**^∆IEC^* mice, thereby confirming that epithelial STAT6 activation is not sufficient to affect Paneth cell differentiation. It is worth noting that, compared with WT mice, TgS6vt mice had a reduced villus length and crypt depth probably resulting from decreased crypt cell proliferation. Notably, we found that STAT6 activation could not reduce the abnormally expanded intestinal crypt size in *Ogt^i^**^∆IEC^* mice, suggesting that OGT regulates crypt cell proliferation through a STAT6-independent manner. However, whether and how OGT-mediated O-GlcNAcylation participates in the regulation of intestinal stem cell and progenitor cell proliferation remain largely unknown.

In summary, we show that the inducible deletion of *Ogt* in IECs of adult mice causes multiple epithelial abnormalities including a reduced number of tuft and goblet cells, the aberrant localization of Paneth cells, and increased crypt cell proliferation. IEC-specific overexpression of constitutively activated STAT6 rescued the tuft and goblet cell differentiation defects caused by *Ogt* ablation. Our findings provide direct in vivo evidence that STAT6 acts as a downstream target of OGT to control type 2 mucosal immune responses.

## 4. Materials and Methods

### 4.1. Mice

*Ogt* floxed mice and Vil-CreERT2 mice were kindly provided by Dr. Xiaoyong Yang (Yale University, New Haven, CT, USA) and Dr. Ye-Guang Chen (Tsinghua University, Beijing, China), respectively. TgS6vt mice expressing STAT6vt (a constitutively activated form of STAT6) driven by the mouse *Vil1* gene promoter were generated by GemPharmatech (Nanjing, China). *Ogt^Ctrl^* and *Ogt^i∆IEC^* mice were generated by crossing *Ogt* floxed mice and Vil-CreERT2 mice. *Ogt^i∆IEC^* mice were bred with TgS6vt mice to generate *Ogt^i∆IEC^*-TgS6vt mice. To induce recombination, 6–8-week-old mice were injected intraperitoneally with tamoxifen (TAM) in corn oil (1 mg/10 g body weight) for 3 consecutive days. Mice were sacrificed 10 days after the final tamoxifen injection and jejunums were harvested for processing. Mice were housed in an environmentally controlled facility with 12 h light/dark cycles, fed standard rodent chow, and had free access to water. All instances of animal use and experiments were approved by the Institutional Animal Care and Use Committee of Xinxiang Medical University, Xinxiang, China. All animal procedures were performed in accordance with “Guide for the Care and Use of Laboratory Animals” published by the National Institutes of Health, Bethesda, MD, USA.

### 4.2. Intestinal Epithelium (IECs) Isolation

Freshly harvested jejunum (~3 cm) was flushed with ice-old PBS, opened longitudinally, and cut into ~0.5 cm pieces. The tissues were rotated at 4 °C in buffer A (3 mM EDTA, 2 mM DTT in PBS) for 15 min, followed by buffer B (3 mM EDTA in PBS) for 45 min. The tissues were then vigorously shaken to release the epithelium, and the supernatants were collected and pelleted for protein or RNA extraction.

### 4.3. RT-qPCR

Total RNA was isolated from jejunal IECs using TRIzol reagent (Takara, Dalian, China) and cDNA was synthesized using a cDNA synthesis kit (Vazyme, Nanjing, China). RT-qPCR reactions were performed using the SYBR Green Master Mix (Vazyme) in an ABI StepOnePlus^TM^ Real-Time PCR system. The relative changes in gene expression were calculated using the ΔΔCt method and normalized to the expression levels of the housekeeping gene *Rplp0*.

### 4.4. Western Blot

Jejunal IECs were lysed in a RIPA buffer containing proteinase inhibitors and an OGA inhibitor. Protein extracts were separated on an SDS-PAGE gel, transferred to nitrocellulose membranes, and blotted with the indicated primary antibodies: OGT (ab96718, Abcam, Shanghai, China), O-GlcNAc (ab2739, Abcam), α-Tubulin (11224-1-AP, Proteintech, Wuhan, China), P-STAT6 (Y641) (ab263947, Abcam), and STAT6 (51073-1-AP, Proteintech). After incubation with HRP-conjugated secondary antibodies, the immune complexes were detected using the ECL detection reagents (Beyotime, Shanghai, China).

### 4.5. Histological Analysis

Jejunal tissues were coiled into “Swiss rolls” and fixed in 4% paraformaldehyde, embedded in paraffin, and sectioned at 5 µm. Paraffin-embedded jejunal sections were stained with H&E according to standard procedures. Alcian blue staining was performed using an Alcian blue stain kit (Abcam) following the manufacturers’ instructions. Immunohistochemistry (IHC) analyses for DCLK1, Ki67 and Lysozyme were performed using an IHC detection kit (ZSGB Bio, Beijing, China) following the manufacture’s guidelines. Alkaline phosphatase (ALP) staining was carried out using the ALP staining kit (Solarbio, Beijing, China), following manufacture’s protocol.

### 4.6. Helminth Infection

Mice were orally infected with 200 *H. polygyrus* L3 larvae with oral gavage, and on day 14 post-infection mice were sacrificed for intestinal tissues collection and for worm burden analysis. For fecal egg counting, fecal pellets were loosened in PBS and eggs were counted using a McMaster counting chamber. The number of eggs was normalized to the weight of feces. For determination of the adult worm burden, the duodenums were cut open longitudinally, followed by incubation in 37 °C PBS for 2 h to harvest and count worm numbers using a dissection microscope.

### 4.7. Statistical Analysis

All data are presented as mean ± SEM. An unpaired Student-*t* test was used to compare the differences between genotypes. *p* < 0.05 was considered to determine statistically significant differences.

## Figures and Tables

**Figure 1 ijms-23-11137-f001:**
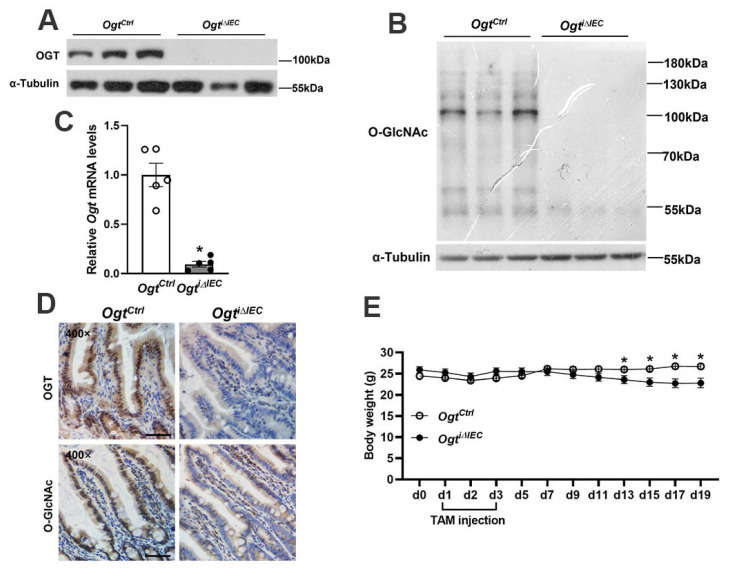
Generation of inducible IEC-specific *Ogt* knockout mice. The 6–9-week-old male *Ogt^Ctrl^* and *Ogt^i^**^∆IEC^* mice were subjected to the following assays. (**A**,**B**) Western blot analysis of OGT (**A**) and O-GlcNAcylation (**B**) in the jejunal IECs. (**C**) qPCR analysis of *Ogt* expression in the jejunal IECs (*n* = 5/group). (**D**) IHC staining for OGT and O-GlcNAcylation in the jejunum. (**E**) Body weight was monitored for a period of 20 days (*n* = 8–9/group). Data are presented as mean ± SEM. * *p* < 0.05. Scale bars, 50 µm.

**Figure 2 ijms-23-11137-f002:**
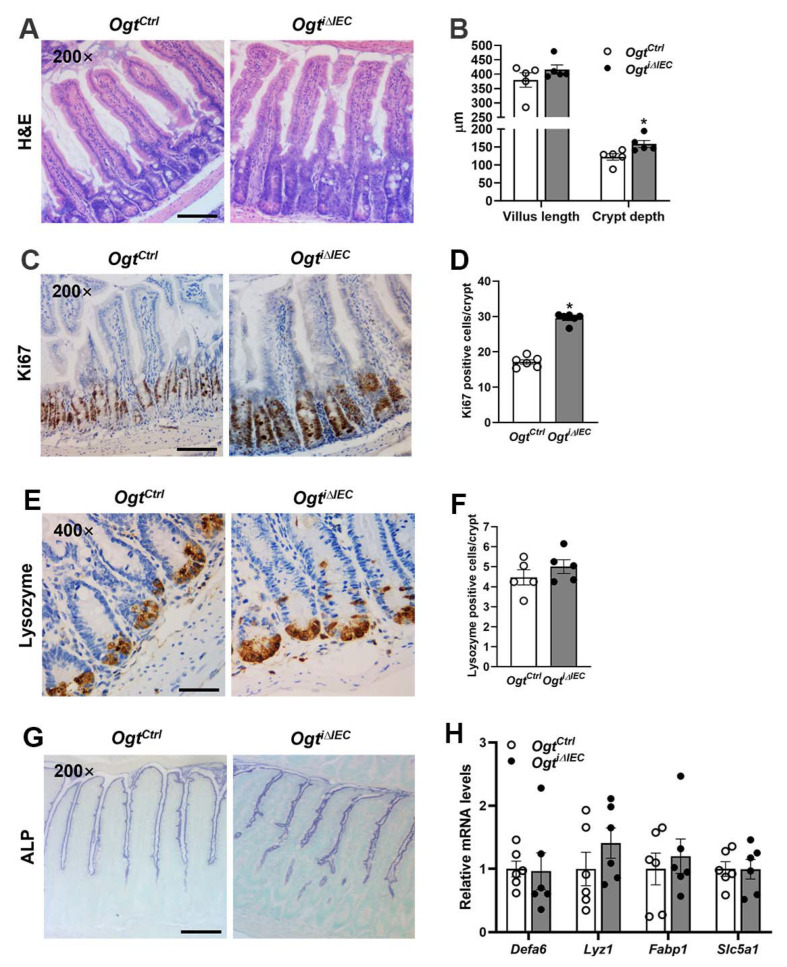
Deletion of *Ogt* in IECs of adult mice leads to abnormal epithelial architecture. The 8–10-week-old male *Ogt^Ctrl^* and *Ogt^i^**^∆IEC^* mice were subjected to the following analyses. (**A**) Representative H&E images of the jejunum. (**B**) Analyses of jejunal villus length and crypt depth (*n* = 5/group, 30 villi or crypts counted for each mouse). (**C**) IHC staining for Ki67 in the jejunum. (**D**) Quantification of Ki67 positive cells shown in (**C**) (*n* = 6/group, 20 crypts counted for each mouse). (**E**) Jejunal Paneth cells were examined by Lysozyme IHC staining. (**F**) Quantification of Paneth (Lysozyme positive) cells shown in (**E**) (*n* = 5/group, 20 crypts counted for each mouse). (**G**) Alkaline phosphatase (ALP) staining of the jejunum. (**H**) qPCR analysis of Paneth (*Defa6*, *Lyz1*) and enterocyte (*Fabp1*, *Slc5a1*) cell markers expression (*n* = 6/group) in the jejunal IECs. Data are presented as mean ± SEM. * *p* < 0.05. Scale bars, 100 µm in (**A**,**C**,**G**); 50 µm in (**E**).

**Figure 3 ijms-23-11137-f003:**
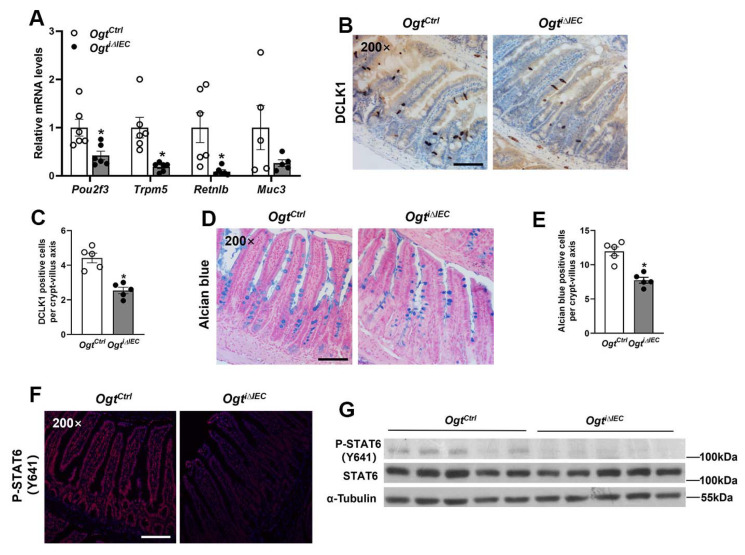
Epithelial OGT is required for tuft and goblet cell differentiation in adult mice. The 8–10-week-old *Ogt^Ctrl^* and *Ogt^i^**^∆IEC^* male mice were subjected to the following analyses. (**A**) qPCR analysis of tuft (*Pou2f3*, *Trpm5*) and goblet (*Retnlb*, *Muc3*) cell markers expression (*n* = 5–6/group). (**B**) Jejunal tuft cells were examined by DCLK1 IHC staining. (**C**) Quantification of tuft (DCLK1 positive) cells shown in (**B**) (*n* = 5/group, 20 crypt-villus units counted for each mouse). (**D**) Jejunal goblet cells were examined by Alcian blue staining. (**E**) Quantification of goblet (Alcian blue positive) cells shown in (**D**) (*n* = 5/group, 20 crypt-villus units counted for each mouse). (**F**) Immunofluorescence for P-STAT6 (Y641) in the jejunum. (**G**) Western blot analysis of P-STAT6 (Y641) expression in the jejunal IECs. Data are presented as mean ± SEM. * *p* < 0.05. Scale bars, 100 µm.

**Figure 4 ijms-23-11137-f004:**
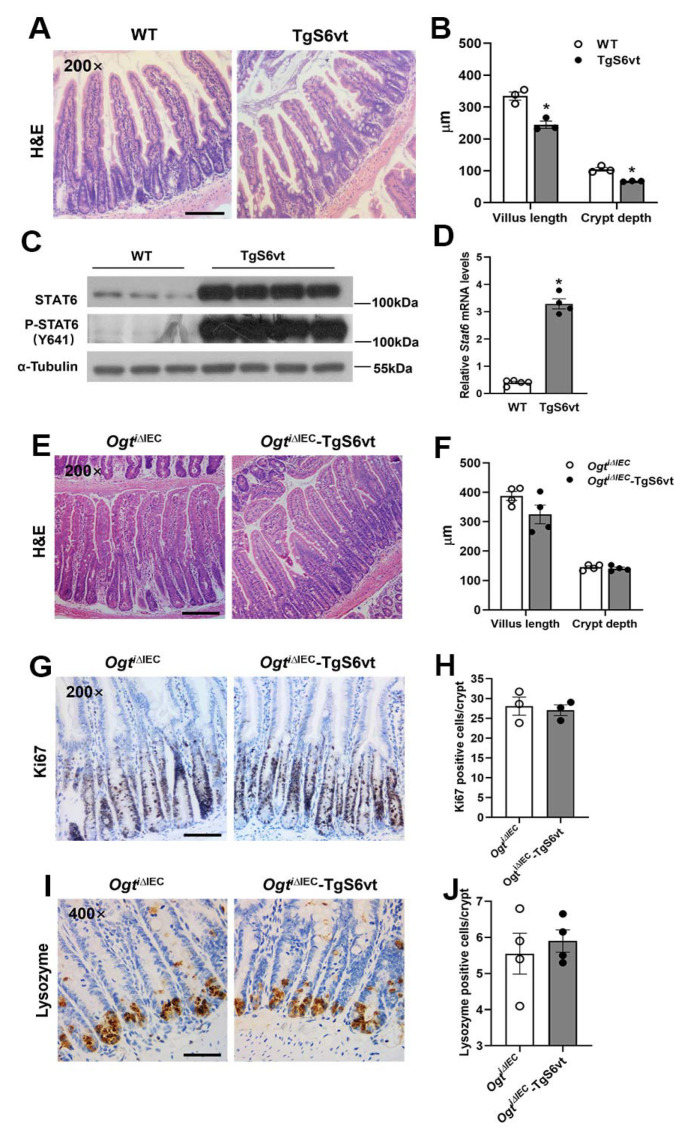
OGT regulates intestinal crypt cell proliferation and Paneth cell differentiation in a STAT6-independent manner. (**A**–**D**) The 7–8-week-old male WT and TgS6vt mice were subjected to the following assays. (**A**) Representative H&E images of the jejunum. (**B**) Analyses of jejunal villus length and crypt depth (*n* = 3/group, 30 villi or crypts counted for each mouse). (**C**,**D**) Western blot (**C**) and qPCR (**D**) analyses of STAT6vt expression in the jejunal IECs (*n* = 4/group). (**E**–**J**) The 9–11-week-old male *Ogt^i^**^∆IEC^* and *Ogt^i^**^∆IEC^*-TgS6vt mice were subjected to the following analyses. (**E**) Representative H&E images of the jejunum. (**F**) Analyses of jejunal villus length and crypt depth (*n* = 4/group, 20 villi or crypts counted for each mouse). (**G**) IHC staining for Ki67 in the jejunum. (**H**) Quantification of Ki67 positive cells shown in (**G**) (*n* = 3/group, 30 crypts counted for each mouse). (**I**) Jejunal Paneth cells were examined by Lysozyme IHC staining. (**J**) Quantification of Paneth (Lysozyme positive) cells shown in (**I**) (*n* = 4/group, 20 crypts counted for each mouse). Data are presented as mean ± SEM. * *p* < 0.05. Scale bars, 100 µm in (**A**,**E**,**G**); 50 µm in (**I**).

**Figure 5 ijms-23-11137-f005:**
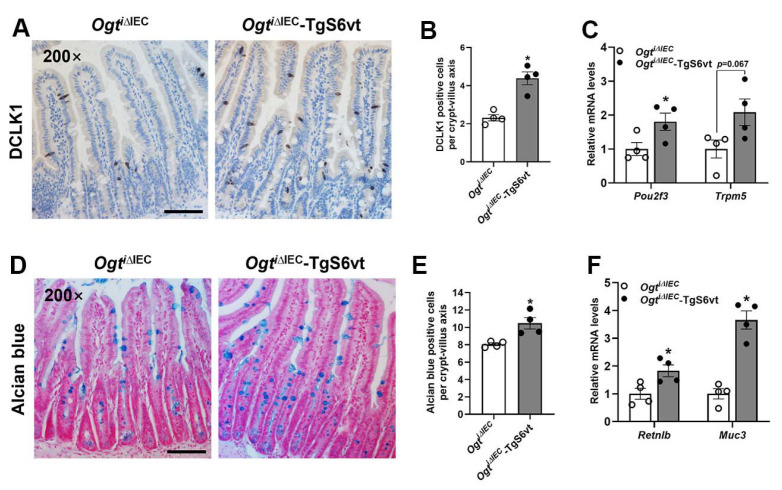
STAT6 activation promotes intestinal tuft and goblet cell differentiation in *Ogt^i^**^∆IEC^* mice. The 9–11-week-old male *Ogt^i^**^∆IEC^* and *Ogt^i^**^∆IEC^*-TgS6vt mice were subjected to the following analyses. (**A**) Jejunal tuft cells were examined by DCLK1 IHC staining. (**B**) Quantification of tuft (DCLK1 positive) cells shown in (**A**) (*n* = 4/group, 20 crypt-villus units counted for each mouse). (**C**) qPCR analysis of tuft cell markers expression in the jejunal IECs (*n* = 4/group). (**D**) Jejunal goblet cells were examined by Alcian blue staining. (**E**) Quantification of goblet (Alcian blue positive) cells shown in (**D**) (*n* = 4/group, 20 crypt-villus units counted for each mouse). (**F**) qPCR analysis of goblet cell markers expression in the jejunal IECs (*n* = 4/group). Data are presented as mean ± SEM. * *p* < 0.05. Scale bars, 100 µm.

**Figure 6 ijms-23-11137-f006:**
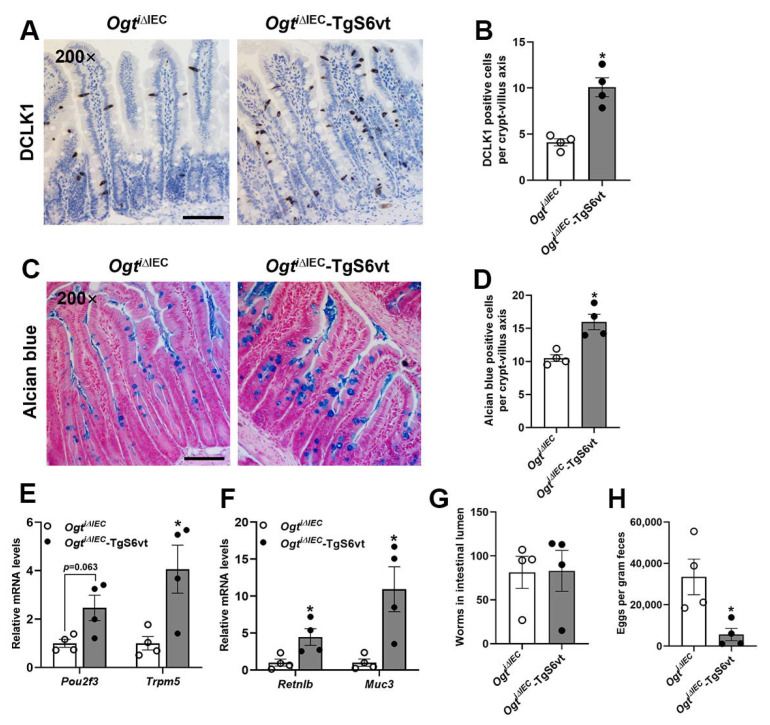
STAT6 overexpression rescues the defective anti-helminth responses caused by epithelial *Ogt* ablation. The 8–10-week-old male *Ogt^i^**^∆IEC^* and *Ogt^i^**^∆IEC^*-TgS6vt mice were infected with *H. poly* and subjected to the following analyses on day 14 post-infection. (**A**) Jejunal tuft cells were examined by DCLK1 IHC staining. (**B**) Quantification of tuft (DCLK1 positive) cells shown in (**A**) (*n* = 4/group, 20 crypt-villus units counted for each mouse). (**C**) Jejunal goblet cells were examined by Alcian blue staining. (**D**) Quantification of goblet (Alcian blue positive) cells shown in (**C**) (*n* = 4/group, 30 crypt-villus units counted for each mouse). (**E**,**F**) qPCR analysis of tuft (**E**) and goblet (**F**) cell markers expression in the jejunal IECs (*n* = 4/group). (**G**,**H**) Analysis of parasite burden by quantification of adult worms in intestinal lumen (**G**) and eggs in feces (**H**) (*n* = 4/group). Data are presented as mean ± SEM. * *p* < 0.05. Scale bars, 100 µm.

## Data Availability

Data supporting reported results can be obtained from the corresponding author under reasonable request.

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
