# Peer review of "Intestinal Epithelial STAT6 Activation Rescues the Defective Anti-Helminth Responses Caused by Ogt Deletion"

_ijms, 2022, doi:10.3390/ijms231911137_

Round 1
Reviewer 1 Report
This work is a continuation of their publication in Immunity earlier this year. It is a well-written article with significant findings. The following concerns were noted.
1. What happens to other parts of the intestine (Ileum and colon) after ogt deletion and stat6 over-expression? Why did the authors select the jejunum for these studies?
2. Does anything happen to deep crypt secretory cells in these animals? (reg4+ cells)
3. The authors state that there were no difference in enterocyte markers in these animals How about important ion transporters such as NHE3 and DRA? Also what happens to the tight junction proteins in these mice.
4. Do these mice show any alterations in type 2 immune cells in the lamina propria or spleen? please discuss.
5. What is the cause of the paneth cell mislocalization? Are these cells observed in the colon as well?
Reviewer 2 Report
The role of O-GalNac glycosylation in immune and other cells including epithelial cells is receiving considerable attention. Activity of this pathway modulates many activities in many cell types and does so by multiple pathways. The present study focuses on OGT activity in intestinal epithelial cell reaction to helminthic infections. Both immune cell functions of the IEC are activated and many demonstrated to be dependent on activation of STAT6 as nicely demonstrated through both deletion by cell specific promoter (villin) Cre recombinase activity and specific overexpression. The effect of helminthic infection on epithelial cell proliferation is demonstrated to occur as in other studies and is not STAT6 dependent.
The manuscript is well written, conservative, and clear. The background includes new concepts on tuft cells that was impressive. Discussion of the diversity of the physiological effects possibly linked to OGT was good. Their use of villin Cre recombinase deletion and specific expression systems to elucidate cell type specific events was good. The in vivo model is a strength of the studies. This studies advance our understanding of the role of OGT in the intestinal response to helminthic infections.
Author Response
Thank you. We appreciate the reviewer very much for his positive comments on our manuscript.